# Shrinkable Hydrogel-Enhanced Biomarker Detection with X-ray Fluorescent Nanoparticles

**DOI:** 10.3390/nano12142412

**Published:** 2022-07-14

**Authors:** Yiting Zheng, Ruiqing Huo, Ming Su

**Affiliations:** Department of Chemical Engineering, Northeastern University, Boston, MA 02115, USA; zheng.yit@northeastern.edu (Y.Z.); huo.ru@northeastern.edu (R.H.)

**Keywords:** nanoparticles, biomarker detection, hydrogel, volume reduction, X-ray fluorescence

## Abstract

This paper reports a new method to enhance the sensitivity of nanoparticle-based protein detection with X-ray fluorescence by exploiting the large volume reduction of hydrogel upon dehydration. A carboxylated agarose hydrogel with uniaxial microchannels is used to allow rapid diffusion of nanoparticles and biomolecules into the hydrogel and water molecules out of the hydrogel. Carboxylated hydrogels are modified to capture protein biomarkers and X-ray fluorescence nanoparticles (iron oxide nanoparticles) are modified with antibodies that are specific to protein biomarkers. The presence of protein biomarkers in solution binds the nanoparticles on the hydrogel channels. The dehydration of hydrogels leads to a size reduction of over 80 times, which increases the number of nanoparticles in the interaction volume of the primary X-ray beam and the intensity of characteristic X-ray fluorescence signal. A detection limit of 2 μg/mL for protein detection has been established by determining the number of nanoparticles using X-ray fluorescence.

## 1. Introduction

Nanoparticles with unique physical properties are promising probes for in vitro detection of biomarkers, antigens, proteins, and nucleic acids [1,2,3,4,5,6]. By converting biological recognition events into physical signals that can be amplified, nanoparticle-based methods achieve high sensitivity, owing to intimate contact with biomarkers in solution. However, most of these methods (except magnetic nanoparticles that can be enriched before readout) require the state-of-the-art electronic and photonic devices to readout the physical signals. The fundamental limit is that nanoparticles are immobilized as a monolayer on a surface in sandwich configuration, where the total number of nanoparticles residing in the interaction volume of incoming electromagnetic excitations will be low [7,8]. If the area density (or the total number) of nanoparticles in the interaction volume could be increased after nanoparticles are captured on the substrate, the intensity of the spectroscopic signal emitted from the nanoparticles will be enhanced without increasing the level of excitation and the sensitivities of instruments [9,10].

Hydrogels that have intrinsic 3D constructs, rich chemistry, and a large surface area are good candidates for biosensing applications [11,12,13,14,15]. Upon dehydration, hydrogels can experience ten- or hundred-fold changes in volume [16,17]. The large volume change ability of hydrogels has been used to enhance the sensitivity of metal ion detection using turn-on organic fluorophores [18]. Ideally, target biomarkers and capturing probes can be attached on hydrogels with high binding capacity (compared to flat configuration) [19,20]. An issue associated with a biosensing hydrogel is that interlaced networks with random pores significantly decrease the diffusivity and permeability of molecules especially biomolecules and large nanoparticle probes [21,22].

This article reports a new method to enhance the sensitivity of nanoparticle-based protein detection using volume-shrink hydrogels, in which X-ray fluorescence nanoparticles are captured on a hydrogel scaffold with an ordered channel structure (Figure 1). Instead of increasing the number of probes to obtain a strong signal, this method will have higher detection sensitivity by concentrating nanoparticle probes in detection volume. The original agarose is activated to carboxylated agarose (CA) by oxidizing its primary alcohol groups before formation of ordered hydrogel scaffold by freeze-thaw method. The aligned channels are produced through temperature gradient controlled ice crystallization and extend the full length of a hydrogel thin film [23,24]. The channels allow rapid diffusion of water, nanoparticles, and proteins. The surfaces of the hydrogels are charged as capturing agents and the surfaces of nanoparticles are modified with the antibodies. In the presence of target antigens, the nanoparticles are immobilized onto the hydrogel through specific antigen-antibody interactions. After rinsing away unbound nanoparticles, the hydrogel is dehydrated to decrease its volume and the attached nanoparticles are enriched to a small volume, which is followed by exposing nanoparticles to an incoming X-ray irradiation and detecting characteristic X-ray fluorescence of nanoparticles [25,26,27,28]. The nanoparticles (iron oxide) are chosen for relatively high X-ray fluorescence yield and readiness of being removed with a magnet if they are not bound on hydrogel.

## 2. Materials and Methods

### 2.1. Chemicals and Materials

Disuccinimidyl suberate (DSS), biotin-PEG_2_-amine and avidin were obtained from Thermofisher (Waltham, MA, USA). Phosphate buffered saline (PBS) was obtained from VWR (Radnor, PA, USA) and Tween 20 was from obtained from Acros (Geel, Belgium). Bovine serum albumin (BSA), dimethyl sulfoxide (DMSO, 99%) and perfluorooctyltrichlorosilane (PFTOS, 97%) were obtained from AlfaAesar (Tewksbury, MA, USA). The following chemicals were from Sigma-Aldrich (Burlington, MA, USA): tetramethyl-1-piperidinyloxy (TEMPO, 98%), (3-aminopropyl) triethoxysilane (APTES), sodium hydroxide (NaOH), sodium bromide (NaBr, 99.0%), sodium hypochlorite solution (NaClO, 10–15%), and iron oxide nanoparticles (25 nm diameter, 5 mg/mL). Agarose powder was obtained from IBI Scientific (Dubuque, IA, USA).

### 2.2. Chemical Modification of Agarose

Agarose was treated with TEMPO-NaBr-NaClO to oxidize the primary alcohol groups to carboxylated agarose (CA) as follows. First, 1 g of agarose powder was dissolved in 80 mL of water at 90 °C. After adjusting the pH to 11 with 1 M of aqueous NaOH, the solution was cooled to room temperature while stirring. Then, 22 mL of an aqueous solution containing 0.02 g TEMPO, 0.3 g NaBr, and 2 mL NaClO was added dropwise into the agarose solution, and the pH of the solution was maintained at 10–11 by adding a 1 M NaOH solution. The oxidation reaction was completed in 1.5 h. The solution was precipitated by adding a mixture of 200 mL of isopropyl alcohol and 20 mL of acetone. The precipitate (i.e., oxidized agarose) was washed twice with ethanol, dialyzed for 48 h, and lyophilized to remove solvents.

### 2.3. Preparation of Hydrogels with Aligned Channels

A freeze-thaw method was used to make hydrogels with vertical channels. First, 0.05 g of carboxylated agarose (CA) and 0.05 g of agarose were dissolved in 10 mL of water at 100 °C, followed by gelation at room temperature. The hydrogel was made into a cylinder that was 1.5 cm in diameter and cut to 2.5 cm in length. The cylinder was placed on a copper block which was positioned at the top of a foam box. The freezing process started when liquid nitrogen was filled in the box, creating a uniaxial thermal gradient. The samples were allowed to freeze for 10 min and removed from the copper block to thaw at room temperature, followed by cutting into thin slabs that were 1.5 mm thick. Fluorescent images of the hydrogels were taken with an Olympus BX51 fluorescence microscope (Center Valley, PA, USA) and scanning electron microscopy (SEM) images were taken with a high-resolution field emission SEM (Hitachi S-4800, Santa Clara, CA, USA).

### 2.4. Biotin-Conjugation on Iron Oxide Nanoparticles

A 0.2 mL aqueous solution of iron oxide nanoparticles was mixed with 5 mL of alcoholic solution of APTES (0.2% by weight) for 2 h under stirring. After removing excess chemicals by centrifugation, the nanoparticles were re-suspended in 2 mL of DMSO containing 4 mg of DSS as a crosslinker for 1 h at 37 °C. The solution was then added into a 2 mL aqueous solution of biotin-PEG2-amine that contains 10 mg of biotin-PEG_2_-amine for 2 h. The biotin modified nanoparticles were rinsed water and centrifuged twice. Attenuated total reflectance Fourier transform infrared (ATR-FTIR) spectra of nanoparticles were collected using a Thermo Scientific Nicolet iS10 FTIR spectrometer (Waltham, MA, USA).

### 2.5. Protein Detection with X-ray Fluorescence

The carboxylated agarose was immersed in avidin solutions (concentrations ranging from 0.2 μg/mL to 2 mg/mL) for an hour at 37 °C and rinsed with 0.1% Tween 20 and water twice. The carboxyl groups on carboxylated agaroses were blocked by immersion in an aqueous solution of BSA (0.5% by weight) for 1 h at 37 °C, followed by rinsing with a Tween 20 solution and water. In order to image the avidin modified hydrogel, iron oxide nanoparticles with biotin labels were added into the hydrogels for 30 min. The excess nanoparticles were removed from Tween 20 solutions using a magnet. The hydrogel was then placed on a PFTOS-modified glass slide and dehydrated in an oven at 50 °C. A Mini-X X-ray tube (Amptek, Bedford, MA, USA) operating at 40 kV and 15 A was used to generate the primary X-ray. The tube was fitted with a brass collimator to reduce the beam size to 1 mm in diameter. An X-ray spectrometer (Amptek X-123, Amptek, Bedford, MA, USA) with Si-PIN photodiode was used to collect X-ray fluorescence (XRF) signals in reflection modes, where the tube and the detector were fixed at an angle of 45° between them. The tube-sample distance and sample-detector distance are kept at 3 and 2 cm, respectively. The whole setup was enclosed in a lead-containing-acrylic chamber. 

## 3. Results and Discussions

Figure 2 shows the details and outcome of hydrogel dehydration. The hydrogel was made as a cylinder that was 1.5 cm in diameter and 2.5 cm in height. The cylinder was placed on a copper block which sat on top of a foam box before immersion in liquid nitrogen (Figure 2A). The construct of the carboxylated hydrogel has been colored by Rhodamine 6G and checked with a fluorescent microscope. Figure 2B shows that the hydrogel has uniaxial channels along the length with an average width of 150 μm. The hydrogel is lyophilized and imaged by SEM (Figure 2C), which shows the channel width in the construct is approximately 120 μm. The 20% reduction in channel width is likely due to volume shrinkage during lyophilization.

The diffusivity of a molecule in the hydrogel was derived by observing the distance of the front of a fluorescent molecule with time while partially immersing the hydrogel in the solution of fluorescent molecules. Figure 3A shows the diffusivities of rhodamine 6G and rhodamine 6G labeled BSA, where (1) rhodamine 6G shows higher diffusivity on hydrogel with random pores (porous hydrogel) and hydrogel with channel structures (construct) than BSA due to smaller size of rhodamine 6G compared to BSA; (2) both rhodamine 6G and BSA show higher diffusivity in hydrogels with channel structures than those with random pores. Figure 3B shows the optical images of the hydrogels before (left) and after (right) dehydration, where the hydrogel changes from transparent to opaque with an 80 times volume reduction. Figure 3C shows the fluorescence images of cross section areas of a dehydrating hydrogel taken at three time points, where the channels in the hydrogel become smaller as dehydration progresses. Figure 3D shows the mass ratios of hydrogels and dehydrated constructs as a function of time. As water evaporates from the hydrogel surface and liquid-vapor interface moves inwards, the diffusion length of water becomes longer, which leads to lower evaporation rate. The dehydration kinetics indicate that the CA-modified hydrogel with vertical channels (black line) dehydrates faster than CA-modified hydrogel with random pores (blue) and native agarose hydrogel with random pores (red).

The iron oxide nanoparticles have been modified to feature biotin at the outmost region. Figure 4A shows the ATR-FTIR spectra collected after modification using pristine nanoparticles (black) as a reference. The APTES-modified nanoparticles show characteristic C-N stretching vibration of amine on APTES at 1275 cm^−1^ (green). DSS covalently immobilized on APTES shows the vibrational peak of C=O from its free NHS ester at 1725 cm^−1^ (blue). ATR-FTIR spectrum also shows the attenuation of C=O vibrational peak at 1725 cm^−1^ after covalent binding of NHS ester of DSS to amine of biotin-PEG-amine (red). In order to have maximal exposure of nanoparticles to incoming X-ray, the cross section of the incident X-ray beam should match the size of the hydrogel construct. The size of a cone shaped X-ray beam is controlled by changing the distance from the X-ray tube and the sample. Figure 4B (inset) shows an optical image of a round fluorescent spot (diameter of 2.5 mm) produced on an X-ray sensitive film placed at 2.5 cm from the X-ray tube. Figure 4B shows the relationship between the spot size of the X-ray beam and the distance, where the spot size decreases as the distance decreases. Figure 4C (inset) shows an XRF peak of iron oxide nanoparticles and the corresponding baseline, which was collected at an accumulation time of 100 s at a distance of 2.5 cm between the X-ray tube and the sample. The oxygen peak at the lower energy side cannot be distinguished. Figure 4C shows the spectra of nanoparticles (red), dehydrated hydrogel (black) and dehydrated hydrogel with nanoparticles (blue), respectively, where the *K**_α_* peak at 6.40 keV and *K**_β_* peak at 7.06 keV of iron element are identified with an area ratio of 6.6:1. Dehydrated hydrogel only shows noise, which is produced by scattering of X-ray on rough hydrogel surface.

Iron oxide nanoparticles were modified with biotin and then used to detect avidin in channeled hydrogels. After immobilization of the nanoparticles on the hydrogel, the intensity of the XRF peak of iron is used to derive the number of attached nanoparticles within the detection volume. Figure 5A shows the relationship between the intensity of detected iron signal before (black) and after (blue) dehydration as a function of hydrogel thickness, which shows the number of nanoparticles increases as the thickness of hydrogel increases in both hydrated form and dehydrated construct. A six-fold increase in the number of detected nanoparticles has been found in a fully dehydrated sample compared to hydrated one, which confirms the sensitivity enhancement due to hydrogel volume reduction. The time it takes for the diffusion of nanoparticles is found nearly identical as the thickness increases from 1.0 to 3.5 mm. Therefore, in the following experiment, the thickness of hydrogel was set to 1.5 mm. A magnet was used to remove unbound nanoparticles prior to hydrogel dehydration. Figure 5B shows the intensities of iron peaks of nanoparticles in dehydrated constructs after detecting avidin at concentrations ranging from 2 mg/mL to 2 μg/mL, where a linear relationship exists between iron peak intensity and avidin concentration with a regression coefficient R^2^ of 0.958. By extrapolating the linear relation (Figure 5B inset), the limit of the detection has been determined to be 2 μg/mL when the nanoparticle probes are enriched in the shrunken hydrogels. A logarithmic scale curve separates the datapoints more clearly at lower concentrations when showing the experimental results in Figure 5B.

## 4. Conclusions

The sensitivity of protein detection with nanoparticle-based X-ray fluorescence (XRF) has been enhanced significantly without the need of sophisticated devices by taking advantage of the large volume reduction potential of hydrogels upon dehydration. Carboxylated agarose hydrogels, containing uniaxial microchannels extending through their entire length, allow high diffusivity for biomolecules and nanoparticles and rapid dehydration of hydrogels. Targeted protein biomarkers have been attracted onto hydrogels via the carboxyl groups, which was followed by the immobilization of the nanoparticles conjugated with antibodies. Upon removal of excess nanoparticles, the dehydration of hydrogel scaffolds causes a significant size reduction (80 times) and an enrichment of nanoparticles in the interaction volume of a primary X-ray beam, leading to a six-fold increase in the intensity of characteristic secondary X-ray emission from nanoparticles. The sensitivity of biotin detection using iron oxide nanoparticle-based X-ray fluorescence is enhanced, resulting in a six-fold increase in the contraction range of 2 mg/mL to 2 μg/mL. Shrunken hydrogen-enhanced biomarker detection is straight-forward, easily performed, and consumes a low level of power. Such a method can be used in a variety of fields where biomarker detection is of importance, such as point-of-care diagnosis and environmental toxicity determination. This method also has the potential to detect multiple biomarkers at the same time by using a variety of metallic or oxide nanoparticles that have characteristic X-ray fluorescence emissions.

## Figures and Tables

**Figure 1 nanomaterials-12-02412-f001:**
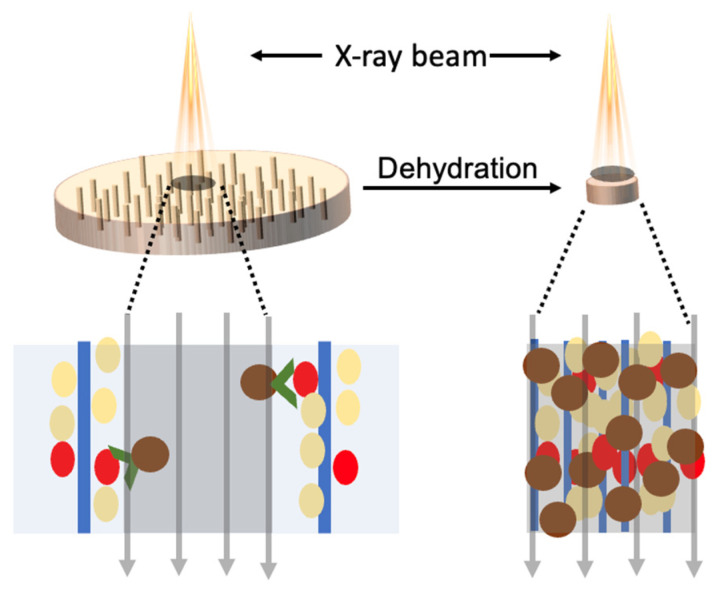
Enhanced X-ray fluorescence detection of biomarkers using nanoparticle probes with dehydrated microchannel hydrogel.

**Figure 2 nanomaterials-12-02412-f002:**
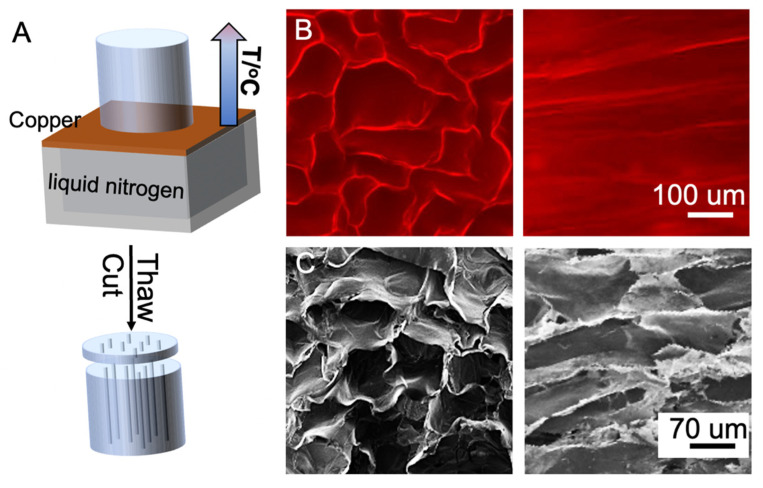
Generation of hydrogels with uniaxial microchannels: (**A**) Fluorescent images (**B**) of a hydrogel at transverse (left) and longitudinal (right) cross sections; SEM images (**C**) of the hydrogel at transverse (left) and longitudinal (right) cross sections after lyophilization.

**Figure 3 nanomaterials-12-02412-f003:**
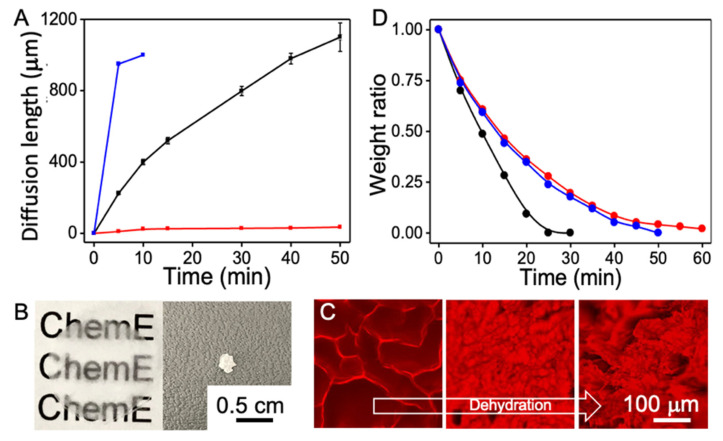
Molecular diffusion in hydrogels: (**A**) BSA in hydrogels with aligned channels (blue), rhodamine 6G in hydrogels with random pores (black), and BSA in hydrogels with random pores (red). Digital images (**B**) of hydrogels before (left) and after (right) dehydration. Fluorescent images (**C**) of hydrogels at transverse direction during dehydration. Weight ratio of water in the hydrogel during dehydration (**D**).

**Figure 4 nanomaterials-12-02412-f004:**
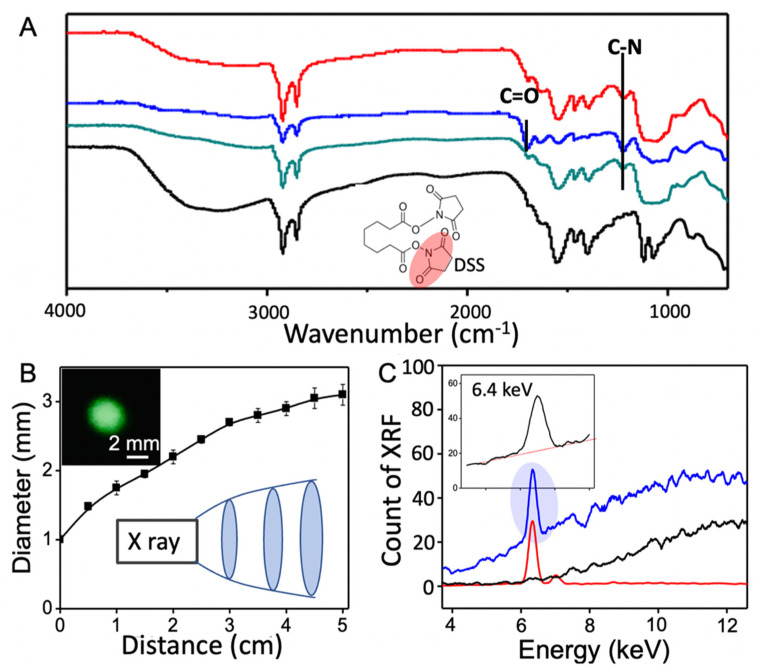
FTIR spectra of nanoparticles after surface modification: (**A**) As obtained nanoparticles (black), APTES modified (green), DSS (blue) and biotin conjugated (red), respectively. The diameter of the primary X-ray spot at various distances (**B**) and an optical image of a fluorescent spot (inset); X-ray fluorescence spectra (**C**) of nanoparticles (red), dehydrated hydrogel (black) and dehydrated hydrogel with nanoparticles (blue), where the inset plot shows the iron peak from nanoparticles in dehydrated hydrogels.

**Figure 5 nanomaterials-12-02412-f005:**
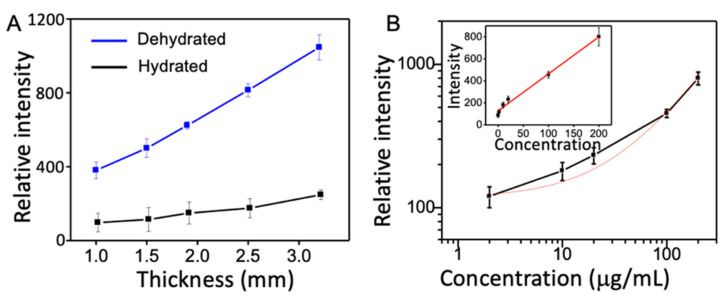
Relative intensity of X-ray fluorescence before (black) and after (blue) dehydration of hydrogels of different thickness (**A**). The relation between X-ray fluorescence intensity and avidin concentration in logarithmic form (**B**) and in linear form (B inset).

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
