# Peer review of "Shrinkable Hydrogel-Enhanced Biomarker Detection with X-ray Fluorescent Nanoparticles"

_nanomaterials, 2022, doi:10.3390/nano12142412_

Round 1
Reviewer 1 Report
This MS deals with a method to increase the diffusivity of NPs in hydrogels characterised by the presence of extensve uniaxial micro-channels.
The technique used, although not entirely new, is interesting and can be used in many biotechnological applications.
The manuscript reads with interest although perhaps a little more detail, especially to defferentiate from other work, would have been helpful.
I have no serious criticisms to make, however some points could be better clarified.
Fig. 3B - the differences between the images of the hydrogels before and after dehydration are not very obvious.
The authors do not explain why the weight ratios of water during the dehydration process are so similar.
Fig 5B - I did not understand which of the two dependencies, the linear and the logarithmic ones, is the correct one.
Finally, the conclusions are just a summary of the results obtained without illustrating what possible applications and improvements the proposed method offers. These aspects should be improved.
Reviewer 2 Report
Authors fabricated carboxylated hydrogel and applied to improve the sensitivity by the dehydrogenation. The fabricated hydrogel was confirmed to be shrunken and to improve the signal intensity of X-ray fluorescence from iron oxide nanoparticle. The topic of this manuscript is well fit to the Nanomaterials and the logic of manuscript is scientifically sound. I would recommend to publish this article as is.
Reviewer 3 Report
The manuscript submitted to Biosensors entitled "Shrinkable Hydrogel Enhanced Biomarker Detection with X-Ray Fluorescent Nanoparticles" by Zheng et al. presents the preparation of a hydrogel combined with iron oxide nanoparticles and Biotin to increase Avidin detection sensitivity.
The overall subject seems interesting, and it might evolve into something useful in the next few years. However, the manuscript discussion looks too simplistic on the overall subject. It isn't easy to understand the importance of using this technique to detect Avidin. The authors refer to Avidin simply as protein throughout the text, making it even more complicated to understand its significance. Instead, the authors should replace protein with the name of the protein.
Furthermore, in line 180, the authors state that iron oxide nanoparticles are modified with Avidin to detect Biotin. During my reading, I was under the impression that iron oxide nanoparticles were modified with Biotin! Please make sure that this is correct.
Overall, the manuscript requires a few clarifications. For instance:
What is the relevance of this detection enhancement and method versus other methods found in the literature for avidin detection?
How does this system compare with comparable systems already published in the literature?
Is the concentration range of 2mg/mL to 2ug/mL relevant to this protein?
Please discuss the answer to these questions in the manuscript.
Thank you
Round 2
Reviewer 1 Report
The authors partly modified their manuscript according to the suggestions of my first report.
The current manuscript, although it could be further improved, nevertheless has reached such a level that it can be published
Reviewer 3 Report
In general, the authors have replied to all my concerns.